# A Psychometric Analysis of the Nurse Satisfaction with the Quality of Care Scale

**DOI:** 10.3390/healthcare10061145

**Published:** 2022-06-20

**Authors:** Bayan Alilyyani, Michael Kerr, Carol Wong, Dhuha Wazqar

**Affiliations:** 1Nursing Department, College of Applied Medical Sciences, Taif University, P.O. Box 11099, Taif 21944, Saudi Arabia; 2Arthur Labatt Family School of Nursing, Western University, 1151 Richmond Street, London, ON N6A 5C1, Canada; mkerr@uwo.ca (M.K.); cwong2@uwo.ca (C.W.); 3Faculty of Nursing, King Abdulaziz University, P.O. Box 24828, Jeddah 21551, Saudi Arabia; dwazger@kau.edu.sa

**Keywords:** satisfaction with the quality of care, quality of care, nursing, psychometric properties, scale, Saudi Arabia, nurses

## Abstract

The concept of quality of nursing care can vary across healthcare organizations, and many different factors may affect the quality of nursing care as perceived by nurses. Measuring satisfaction with quality of nursing care from the nurse’s perspective is important as a valid and reliable indicator of care quality. The purpose of this study was to measure the psychometric properties of a researcher-developed instrument measuring nurse satisfaction with quality of care. A sample of 200 nurses was randomly selected from three different cities in Saudi Arabia and surveyed with the Nurse Satisfaction with Quality of Care Scale, which is a self-administrated five-item scale. Exploratory factor analysis, confirmatory factor analysis, and internal consistency analysis were conducted to assess aspects of the validity and reliability of the instrument. The results of exploratory factor analysis supported a one-factor structure that consisted of the five items. Confirmatory factor analysis results confirmed that the five items were integral to nurse satisfaction with quality of care. The Cronbach internal consistency of the scale was acceptable. The scale appeared to be a reliable and valid tool for assessing nurse perceptions of their satisfaction with the quality of care provided. Additional studies to further test psychometric properties of this scale in different contexts are warranted.

## 1. Introduction

Nurses’ perspectives on quality of patient care they deliver and what is needed to improve that care develop over time as they engage in daily bedside nursing practice [1]. Furthermore, nurses’ perceptions of care quality have been shown to be a reliable indicator of the actual care delivered [2]. Nursing care quality is defined as the care that is given by nurses based on healthcare organizational standards and nursing professional requirements [3]. Different concepts such as nursing skills, effective management and leadership, and effective community participation [4], adequate nursing staff, as well as patient outcomes including patient satisfaction [5] have all been considered as important elements of nursing care quality. According to Burhans and Alligood [6], nurses improve the quality of nursing care based on assessment, intervention, and effectiveness of nursing treatments. Different factors can affect quality of nursing care such as the work atmosphere, equipment shortages and inefficiency, dissatisfaction with the facilities, and poor economic status of the facility [7]. Enhancing the nursing work environment is an essential condition for nurses to provide the quality of care needed for each patient which, in turn, can increase nurse job satisfaction [8,9]. It has been shown that when organizational support including adequate equipment, technological advances and appropriate facilities is provided to nurses, it positively affects nursing care quality [10,11].

Quality of nursing care has been associated with various patient and nurse work outcomes. Research has shown that nurses’ perceptions of patient quality of care are associated with nurse outcomes such as job satisfaction, career satisfaction, and intention to leave [12,13]. Aiken et al. [8] found that, in all of the 12 countries they examined, quality of nursing care was significantly associated with patient satisfaction with care, and nurse workforce outcomes including burnout, dissatisfaction, and turnover. A study conducted by McHugh and Stimpfel [2] reported that the quality of nursing care rated by nurses had a significant relationship with objective outcomes related to hospital quality indicators including death rate and patient satisfaction. Another study explained that an increased quality of nursing care may improve patient and organizational outcomes such as patient falls, infections, and medication errors [14].

Measuring the quality of nursing care by nurses is essential because it helps to determine the level of treatment success and to evaluate care provided to patients [15]. In addition, measuring nurses’ satisfaction with the care is a way to determine how they view the quality of care [8]. Spence Laschinger et al. [1] found that when new graduate nurses perceived they were providing nursing care based on professional standards, their assessment of overall care quality was positively associated with their overall job satisfaction. Nurses can practice and provide care according to professional standards when they work in supportive environments [16]. For example, it was found that if nurses were engaged in their work, it positively affected their perception of quality of care [17]. Thus, nurse satisfaction with the quality of care may be a proxy measure of the actual quality of nursing care delivered [8]. Measuring nurse satisfaction with the quality of care offers an important perspective when assessing quality of nursing care; nonetheless, there is no pre-existing standardized scale for measuring nurse satisfaction with the quality of care.

There is a gap in the nursing literature regarding the measurement of nursing care quality in general; perhaps this is due to the subjective nature of the definition of quality which results in measurement challenges [6]. Defining the quality of nursing care is variable as each nurse may evaluate the quality of care from differing perspectives [6] and the resulting inconsistency in definition of nursing care quality makes it challenging to measure [18,19].

This significant gap might be addressed by focusing instead on measuring the degree of nurses’ satisfaction with the quality of care they provide, as the quality of nursing care is a core concept in nursing practice [8]. Measuring the level of nurse satisfaction with quality of care as an essential element of a work setting could enable nurses who are trying to reach the goals of the organization related to providing high quality nursing care [20]. In addition, nurse satisfaction with the quality of care may have an influence on patient outcomes. For example, nurses working in poor work environments such as insufficient resources and having to care for a high number of patients on their shifts could be more likely to make errors that could lead to negative outcomes related to patients and thus could feel more dissatisfied with the quality nursing care delivered [21]. Although measuring nurse satisfaction with the quality of care is important, previous studies have focused on measuring satisfaction with the quality of care primarily from the patients’ perspective [22,23], and none measured satisfaction with the quality of care from the perspective of nurses.

There is no existing published scale designed to measure nurse satisfaction with the quality of care. Thus, Laschinger and Kerr developed a scale to measure nurse satisfaction with the quality of nursing care provided, although this scale has not yet been published or tested for psychometric analysis. Known as the Nurse Satisfaction with the Quality of Care Scale (NSQC), it consists of five items, rated on a five-point Likert-type scale ranging from 1 (very dissatisfied) to 5 (very satisfied). Laschinger and Kerr (unpublished) developed the items of this scale for use in the National Survey of the Work and Health of Nurses (NSWHN) in Canada as part of the NSWHN’s Perception of Quality of Care section. Laschinger and Kerr developed the quality of care items based on the key quality of nursing care recommendations from The Nursing Sector Study in its Phase II final report. Therefore, the new scale was designed to measure nurse satisfaction with the quality of care they provided, although no other published studies have described the psychometric properties of this scale to date.

Therefore, the goal of this study was to assess the psychometric properties of an unpublished instrument that measures nurse satisfaction with the quality of nursing care. This scale focuses on measuring nurse satisfaction with the quality of care rather than directly measuring objective indicators of nursing care quality. Testing psychometric properties of this scale included exploratory factor analysis (EFA) to assess the need for item reduction, confirmatory factor analysis (CFA) to examine the factor validity of the scale, and Cronbach’s alpha to assess the scales’ internal consistency (i.e., reliability).

## 2. Materials and Methods

### 2.1. Design

This study used a quantitative methodology in order to evaluate the psychometric properties of the Nurse Satisfaction with the Quality of Care Scale.

### 2.2. Sample and Setting

Two hundred nurses were selected to complete the survey used for the psychometric analysis of nurse satisfaction with the quality of care. The sample was drawn from 656 subjects obtained for a larger study that examined the relationships between authentic leadership, job turnover intentions, and satisfaction with the quality of care among nurses who worked in public hospitals in three cities (Makkah, Jeddah, and Taif) in Saudi Arabia. A total of 656 out of the 1130 questionnaires distributed were completed and returned in that study, being a response rate of 58%. Non-probability, convenience sampling was used. The 200 subjects used in this study were randomly selected from the 656 obtained for the larger study, by using SPSS. A forum sample size calculation was not obtained, although 200 subjects can be considered an adequate sample for conducting a factor analysis (Kline, 2005).

The study subjects were recruited from three public hospitals in three different cities in Saudi Arabia, including Makkah, Taif, and Jeddah, for approximately four months during 2019 (May–August). For this methodological sub-study, 200 of the 656 participants from the main study were randomly selected for surveying with the psychometric analysis of Nurse Satisfaction with the Quality of Care Scale.

### 2.3. Inclusion/Exclusion Criteria

Only participants who were registered nurses in Saudi Arabia, worked on inpatient units and outpatient clinics, had six months or more of experience in their current department to ensure familiarity with the setting and their manager, were in direct nursing care positions, were willing to participate in the study, and were capable of completing the survey in English, were included in this study.

### 2.4. Instrumentation

The NSQC scale is a self-administrated 5-item scale designed to measure nurse satisfaction with quality of care that they provide to their patients. Nurses rate their level of satisfaction for each item on the NSQC from 1 (very dissatisfied) to 5 (very satisfied), with higher scores indicating higher levels of satisfaction. A demographic questionnaire was also used to gather descriptive information about the study participants.

### 2.5. Analysis

Two statistical packages were used to analyze data, including the Statistical Package for Social Sciences (SPSS 25) [24] and Mplus 8 [25]. Cronbach’s alpha reliability coefficient was used to determine the scale’s internal consistency. Item loadings and factorial validity of the scale were examined using a two-step factor analysis process. First, the scale’s dimensionality was examined by using exploratory factor analysis to test for redundant items and the appropriate number of factors (using factor loadings higher than 0.35). Second, confirmatory factor analysis was used to test the fit of the data with the goodness-of-fit of the model based on the following criteria: Chi-square (χ^2^), degrees of freedom (df), root mean square error of approximation (RMSEA) (the required score was <0.06), standardized root mean square residual (SRMR) (the required score was <0.08), and comparative fit index (CFI) (the required score was >0.95) [26,27].

### 2.6. Ethical Approval

Ethical approval for the study was obtained from the Saudi Ministry of Health (IPR number HAP-02-T-067, approval number 225) and the Western University Human Research Ethics Board (Project ID 113798).

## 3. Results

### 3.1. Sample Characteristics

Most respondents were female (92%) with a mean age of 32 years (SD = 6.56). The mean of years of experience as registered nurses was nine years (SD = 5.99) and at their current unit was five years (SD = 4.46). Most nurses were not Saudi nationals (including Indian 44.5%, Filipino 26%, other 3.5%) with Saudis accounting for only 25% of the total. Most nurses (68%) had a Bachelor’s of Nursing degree, while 30% held a Diploma in Nursing, and just 2% held a Master’s degree in Nursing. Nurses worked in a variety of different departments such as medical unit 17%, surgical unit 28%, ICU 7%, and cardiac unit 7.5%. Most of the nurses (54%) were from Jeddah, while 26% were from Makkah, and 20% from Taif.

### 3.2. Descriptive Statistics

The mean for the each of the five items from the NSQC ranged from a low of 2.75 for item 3 (satisfaction with the level of staffing that is available for patient care in this unit/clinic) to a high of 3.76 for item 1 (the type of care you can provide to patients in this unit/clinic). The standard deviations were similar for the five items, and there were no outliers observed. Cronbach’s alpha for the five items of the NSQC was 0.80, indicating that the scale was reliable. Table 1 illustrates the descriptive statistics of the five items of the scale.

### 3.3. Exploratory Factor Analysis

The results of the exploratory factor analyses with oblimin rotation showed that a one-factor solution with the five observed variables is recommended for the following reasons. Firstly, the initial Eigen values from the EFA supported a one-factor solution. This factor accounted for 57.58% of the variance in the five items. In addition, the scree plot figure showed that there was only one factor with an Eigen value above 1 (see Figure 1). The results of Kaiser–Meyer–Olkin (KMO) to measure the sampling adequacy (0.74) and Bartlett’s test of sphericity (Chi-Square = 383.26, df = 10, *p* = < 0.001) also supported the one-factor structure.

### 3.4. Confirmatory Factor Analysis

The fit statistics of the CFA model are presented in Table 2. The results of the analysis of the original model of the five observed variables were: χ^2^ (5) = 89.96, *p* < 0.001, RMSEA = 0.29 (90% CI = 0.24–0.34), CFI = 0.77, SRMR = 0.08. Based on the results-of-fit indices, the model did not have an adequate fit. The modification that was made was based on the modification indices suggested by the Lagrange multiplier tests and indicated a need to correlate the error terms between the residuals for item 3 (“the level of staffing that is available for patient care in this unit/clinic”) and item 4 (“the availability of other resources needed for patient care in this unit/clinic”) which suggested that these two items may be theoretically similar [28]. These items could be theoretically similar because they measured the availability of two essential components of the quality of care in the unit which are staff and resources. This makes sense as the availability of staff and resources are considered as essential elements of satisfaction with quality of care in the practice. Additionally, staff and resources are the foundation of the care that each patient needs to receive the best care.

The results of CFA showed there was an error correlation between the residuals of the two items. Thus, allowing for this correlation of the errors between the residuals improved the goodness-of-fit indices χ^2^ (4) = 9.92, *p* = 0.04, SRMR = 0.02, RMSEA = 0.08 (90% CI = 0.01–0.15), CFI = 0.98. The fit of the final model was therefore improved, as evidenced by the fit indices shown in Table 2.

The range of factor loadings was 0.44–0.76. All loading factors were statistically significant at *p* < 0.001. Figure 2 illustrates the factor loadings of the five observed variables model, as well as the correlation between two observed variables residuals for item 3 (the level of staffing that is available for patient care in this unit/clinic) and item 4 (the availability of other resources needed for patient care in this unit/clinic). Their correlation was 0.60 and statistically significant at *p* < 0.001. The correlations between observed variables ranged between 0.33 and 0.70 (see Table 3).

## 4. Discussion

To achieve the goal of this study, exploratory factor analysis and confirmatory factor analysis were used to assess the structural validly of the scale, as well as Cronbach’s alpha to assess the reliability. It is essential for nurses to be aware of the quality of care that they provide for their patients as this helps to maximize their satisfaction with the care and the quality of their patients’ outcomes [22,29]. As previously mentioned, no pre-existing standardized scale was found to measure nurse satisfaction with quality of care. Although the various definitions of quality of care can complicate its assessment, in this study, the focus was on measuring nurses’ perceptions of their satisfaction with the quality of care provided in their unit. By focusing on nurses’ degree of satisfaction with specific elements of quality of care such as the type of care they provide, the time they spend with each patient, the availability of resources and staffing, and overall care provided in the unit, this scale provides more information than a single-item global, or overall quality of care rating could provide.

The range of the mean for each of the five items was 2.75–3.76. The highest mean score was for the item relating to the type of care provided to patients in the unit/clinic, while the lowest mean was for the level of staffing that was available for patient care in the unit/clinic. The results supported previous studies that showed that adequate staffing is a significant issue in some countries such as the United States and Europe [8,30]. It is also considered to be an important issue in Saudi Arabia, where Alharbi et al. [31] found that supporting nurses through adequate staffing could improve the quality of nursing work life in Saudi Arabian hospitals.

The results of this study suggested that the NSQC is a valid and reliable research tool. The results support a one-factor solution that consists of the five items as the best fit for the scale, with all five items correlated to one factor. No item was eliminated, which means that these five items appear to all independently contribute to measuring nurse satisfaction with the quality of care. The range of the loading factors was 0.58–0.74, which are considered acceptable loading values [32].

Assessing the reliability and validity of a research instrument helps future studies intending to use the scale. The importance of measuring nurse satisfaction with quality of care to ensure providing best care has been demonstrated in previous research [30,33]. Because nurses are considered as the largest group who provide healthcare for patients in most healthcare organizations [34], evaluating the quality of care that they provide and assessing their level of satisfaction with nursing care is essential [8]. The five items of this scale summarize what has been found in the literature in terms of the factors that could determine nurses’ level of satisfaction with quality of care. Availability of resources and supplies, adequate staff and organizational support were considered as essential components of providing nursing care with a high level of satisfaction [8,9,10,35]. Other studies mentioned that one of the most significant factors that affect nurse satisfaction and nurses’ evaluations of the quality of care was the work environment and the availability of equipment [36,37]. Tervo-Heikkinen et al. [38] illustrated that the proportion of registered nurses in relation to all staff and their work experience could also affect their overall satisfaction. Thus, the five items of the scale were related to these elements that determined how nurses were satisfied with the type of care, amount of time spending with patients, adequate staff, availability of resources and overall quality of care.

The results of this study suggest that these five items represent a reliable and valid way to measure nurse satisfaction with the quality of care, based on the statistical results and the theoretical approach used. These five items represent the level of nurse satisfaction with the type of care they provide to patients, the amount of time spent with patients, the availability of staff, the availability of resources, and the overall quality of care that is needed to meet patients’ needs regarding to nursing care. The focus of the scale was on patient centered-care, which has been reported to be the prime indicator of quality of nursing care [39]. Therefore, the items of the scale were found to conceptually map onto key measures of nurse satisfaction with the quality of care.

## 5. Conclusions

The findings of this study provide evidence relating to the psychometric properties of a new tool that measures nurse satisfaction with the quality of care. The scale was found to be reliable and valid for use by nurses in hospital-based practices in Saudi Arabia. The scale consists of five items developed to measure nurse satisfaction with the quality of care provided in their practice area. EFA and CFA results provide evidence that this scale measures nurse satisfaction with the quality of care, while the result of Cronbach’s alpha indicated the reliability of the scale. The quality of nursing care has a significant impact on patient, nurse and healthcare organization outcomes, so the benefits of using this scale could extend to hospital management, clinical nursing practice, and future research. This tool is useful not only for nurses and nursing practice, but also for healthcare organizations seeking to improve patients’ outcomes and to evaluate nurses’ satisfaction with the quality of nursing care being provided. There is a need for future studies to use this scale in various contexts so that its psychometric properties and links with observed quality of care can be further established.

## Figures and Tables

**Figure 1 healthcare-10-01145-f001:**
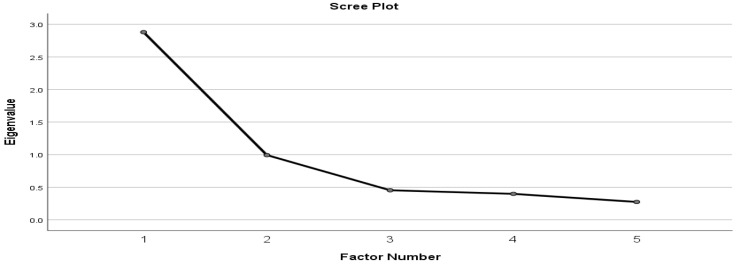
The Scree Plot.

**Figure 2 healthcare-10-01145-f002:**
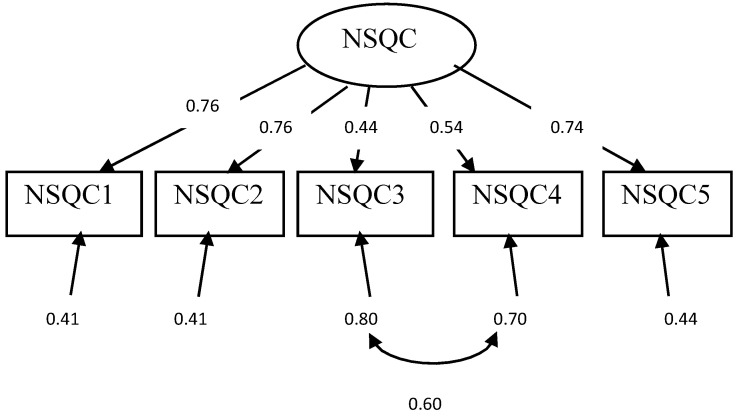
The standardized observed variables loadings model. NSQC, Nurse Satisfaction with Quality of Care. Note: all coefficients were statistically significant (*p* < 0.001).

**Table 1 healthcare-10-01145-t001:** Descriptive statistics of the five items of the scale.

Items	Mean	Std. Deviation	N	Skewness	Kurtosis
1- The type of care you can provide to patients in this unit/clinic.	3.76	0.84	200	−1.08	1.94
2- The amount of time you can spend with patients in this unit/clinic.	3.64	0.85	200	−0.77	0.93
3- The level of staffing that is available for patient care in this unit/clinic.	2.75	1.19	200	0.07	−1.00
4- The availability of other resources needed for patient care in this unit/clinic.	3.04	1.03	200	−0.10	−0.63
5- The overall quality of care patients receive in this unit/clinic.	3.51	0.94	200	−0.62	0.53

**Table 2 healthcare-10-01145-t002:** Summary of original model, and the model modification.

Model	Summary of Modifications	χ^2^ (df)	*p*	RMSEA	CFI	SRMR
Original	N/A	89.96 (5)	0.001	0.29 CI: 0.24–0.34	0.77	0.08
Modification 1	Allowing the correlation of errors between the residuals of items 3 and 4	9.92 (4)	0.04	0.08 CI: 0.01–0.15	0.98	0.02

**Table 3 healthcare-10-01145-t003:** Correlation matrix between observed variables.

Observed Variables	1	2	3	4	5
1. NSQC 1	_	_	_	_	_
2. NSQC 2	0.58 **	_	_	_	_
3. NSQC 3	0.34 **	0.34 **	_	_	_
4. NSQC 4	0.41 **	0.41 **	0.70 **	_	_
5. NSQC 5	0.56 **	0.57 **	0.33 **	0.40 **	_

NSQC, Nurse Satisfaction with Quality of Care; ** *p* < 0.001.

## Data Availability

Data is available on request, due to ethical restrictions.

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
