# Peer review of "A Psychometric Analysis of the Nurse Satisfaction with the Quality of Care Scale"

_healthcare, 2022, doi:10.3390/healthcare10061145_

Round 1
Reviewer 1 Report
Thank you for the opportunity to review this manuscript, an interesting and worthwhile paper for publication. The following are my comments:
Line 36 - re consider this sentence and how you are incorporating community into this sentence. There needs to be a comma after skills.
Line 83 - work do you mean by the term poor work environment needs further explaining
Line 91 - Be careful when commencing a new paragraph with As mentioned before you need to restate what you are going to discuss in this paragraph, be more specific.
Line 114 - please re word and critically consider methodology as quantitative and method as to how you generated the data.
Line 164 - you need to use the word eg nine/five up to nine and then you can use the numerical 10 onwards
Hope this is helpful
Reviewer 2 Report
The authors attempted to construct The Nurse Satisfaction with The 2 Quality of Care Scale. I find their work crowned with success. The newly developed tools may contribute to the improvement of the quality of medical care as well as the satisfaction with the work of the nursing staff. I rate the article submitted for review very highly. With the constructed tool, the authors filled the gap that had so far been the subject of this research. Below are some of my comments for the authors to consider.
Both in the introduction and in the discussion, one could also refer to the following articles:
https://doi.org/10.3390/healthcare10030518
https://doi.org/10.3390/ijerph19127244
https://doi.org/10.3390/ijerph19063620
https://doi.org/10.3390/ijerph19063483
Are there any studies showing a relationship between the subjective perception of quality of care and work engagment? (Thus, nurse satisfaction with the quality of care 66 may be a proxy measure of the actual quality of nursing care delivered [8].)
2.3. Inclusion/Exclusion Criteria
Can the authors give an approximate percentage of nurses who speak English in the general population?
Do the authors believe that the scale can only have 4 items? the correlation between the third and fourth question is very large. Additionally, The Scree Plot shows little gain between 4 and 5
